# Interpreting polygenic score effects in sibling analysis

**Jason Fletcher**[1]*, **Yuchang Wu**[2], **Tianchang Li**[2], **Qiongshi Lu**[3]

1 La Follette School of Public Affairs, University of Wisconsin-Madison, Madison, WI, United States of America, 2 Center for Demography of Health and Aging, University of Wisconsin-Madison, Madison, WI, United States of America, 3 Department of Biostatistics and Medical Informatics, University of Wisconsin-Madison, Madison, WI, United States of America

* jason.fletcher@wisc.edu

**Data Availability Statement:** The data underlying the results presented in the study are available from SPARK (Simons Foundation Powering Autism Research for Knowledge) study https://www.sfari.org/resource/spark/

## Abstract

Researchers often claim that sibling analysis can be used to separate causal genetic effects from the assortment of biases that contaminate most downstream genetic studies (e.g. polygenic score predictors). Indeed, typical results from sibling analysis show large (>50%) attenuations in the associations between polygenic scores and phenotypes compared to non-sibling analysis, consistent with researchers' expectations about bias reduction. This paper explores these expectations by using family (quad) data and simulations that include indirect genetic effect processes and evaluates the ability of sibling analysis to uncover direct genetic effects of polygenic scores. We find that sibling analysis, in general, fail to uncover direct genetic effects; indeed, these models have both upward and downward biases that are difficult to sign in typical data. When genetic nurture effects exist, sibling analysis creates "measurement error" that attenuates associations between polygenic scores and phenotypes. As the correlation between direct and indirect effect changes, this bias can increase or decrease. Our findings suggest that interpreting results from sibling analysis aimed at uncovering direct genetic effects should be treated with caution.

## Introduction

Due to the high correlation in genetic measurement between offspring and parents, it is difficult to separate "direct" genetic effects of offspring genotype on phenotype with "indirect" genetic effects of parental genotype on offspring phenotype. This issue was demonstrated empirically by Kong et al. [1], who showed that associations between non-transmitted parental alleles and offspring phenotype explained approximately 30% of the r-squared of the offspring polygenic scores (PGS) on offspring phenotype in the case of educational attainment.

Researchers have since made the conjecture that sibling models can solve this and other problems, since biological siblings share the same parents and therefore may share the same indirect genetic effects. The more general claim of the usefulness of siblings and the associated "genetic lottery" proceeds the results of indirect genetic effects (see Fletcher and Lehrer [2, 3]). Indeed, many subsequent analyses have demonstrated important reductions in the estimated associations between PGS and phenotypes after controlling for sibling fixed effects, often on

**Funding:** The authors gratefully acknowledge use of the facilities of the Center for Demography of Health and Aging at the University of Wisconsin-Madison, funded by NIA Center Grant P30 AG017266.

**Competing interests:** The authors have declared that no competing interests exist.

the order of 50% (Trejo and Domingue [4], Selzam et al. [5]). This theory and evidence have increased researchers' confidence in interpreting sibling models as producing causal (direct) genetic effects—both in upstream GWAS analysis (Howe et al. [6]) and in downstream PGS analysis (Belsky et al. [7]). For example, in a recent review article, Harden and Koellinger [8] state:

> "Because genotypes are assigned randomly with respect to all other variables, an association between sibling differences in PGS and sibling differences in phenotype is powerful evidence that the PGS is tapping genetic variants with a causal influence on the phenotype."

While intuitive, these interpretations have not been subject to theoretical and empirical scrutiny. We use family data (quads) combined with simulation evidence to show that this intuition relies on very simple models of genotype-to-phenotype associations, where direct and indirect genetic effects are separable (and thus differenced out in sibling models). While researchers have suggested that using a sibling model would purge the indirect effects, we show that, even in our simplified cases, researchers have failed to recognize the dual role of indirect genetic effects as a confounder that produces both positive and negative bias on the estimated effects of offspring PGS. We build the intuition for the negative bias by considering a scenario where all SNP effects are causal and there are no indirect genetic effects and show that sibling analysis leads to attenuated estimates of PGS effects. Essentially, the attenuation stems from sibling analysis' elimination of both confounding and true genetic effects. We then extend this scenario to allow correlation between indirect and direct genetic effects to show that the combination of biases in general can be negative or positive and depend on: the strength of indirect effects, the correlation between indirect and direct genetic effects, and also the extent to which sibling's environments are correlated. In summary, even our simple data generating process shows that sibling analysis is unlikely to produce accurate estimates of direct effects and, worse, does not suggest clear correctives or bounding exercises that are effective.

## Result

The detailed simulation procedures are described in the Materials and methods section. Briefly, we draw the direct and indirect effect sizes for each SNP as well as the environmental effect for each individual following a specific set of parameters in each scenario. Then, we regressed the phenotype constructed by adding up the components above on the theoretical PGS estimation that can be obtained as outputs of genome-wide association studies (GWAS). We compared either the regression coefficient or $R^2$ of both between- and within-family regression to imitate real PGS analysis and investigate the impact of changes of parameters in the data generating process on the performance of sibling PGS analysis. Table 1 provides a summary of the inputs and analysis output across our scenarios below.

### Special case of no indirect genetic effects: The influence of the correlation of environmental effects between siblings

We performed simulations under the simple scenario where the phenotype has no indirect genetic effect contribution ($\sigma_i = 0$); one example phenotype for this scenario could be height, which has been shown to have minimal indirect genetic effects (Kong et al. [1]). In this case, from Formulas (10, 11) in the method section, we can see that sibling PGS models are not guaranteed to estimate the variance component of direct effects even when genetic nurture is absent. This is because the elimination of family effects that occur in sibling models also

**Table 1. Summary of calculations for regression estimates.**

| Study design | Regression model | Regression coefficient | $R^2$ |
|---|---|---|---|
| Between family | $Y \sim PGS_{mix}$ | 1 | $\frac{\sigma^2_{dir}+\sigma^2_{ind}+2\rho_g}{\sigma^2_{dir}+2\sigma^2_{ind}+2\rho_g+\sigma^2_e}$ |
| Sibling difference | $\Delta Y \sim \Delta PGS_{mix}$ | $\frac{\sigma^2_{dir}+\rho_g}{\sigma^2_{dir}+\sigma^2_{ind}+2\rho_g}$ | $\frac{(\sigma^2_{dir}+\rho_g)^2}{(\sigma^2_{dir}+\sigma^2_{ind}+2\rho_g)[\sigma^2_{dir}+2\sigma^2_e(1-r_e)]}$ |
| Sibling difference | $\Delta Y \sim \Delta PGS_{dir}$ | 1 | $\frac{\sigma^2_{dir}}{\sigma^2_{dir}+2\sigma^2_e(1-r_e)}$ |

PGS regression coefficients and $R^2$ in different study designs. Different study designs can be used for PGS regression analysis. In a "between family" analysis, phenotype $Y$ is regressed on PGS. In a sibling difference design, the phenotype difference between siblings is regressed on their PGS difference. PGS can be calculated using either a mixture of direct and indirect SNP effects ($PGS_{mix}$) or using only the direct effect ($PGS_{dir}$).

eliminate true direct genetic effects. Our formulas show that the elimination of these true genetic effects can (only) be offset in the (unlikely) case where siblings have a correlation of 0.5 in their environmental effects ($\rho_e = 0.5$); in this case, the correlation in both environmental and genetic effects are the same. In general, when siblings have a correlation in their environmental effects higher than 0.5, the $R^2$ of sibling PGS model is overestimated; when this correlation is below 0.5, the $R^2$ of sibling PGS model is underestimated (Fig 1). Since the environmental effect is assumed to be independent of genetic components in this study, this component does not interact with other potential factors. Therefore, to focus on the influence of other parameters (and not $\rho_e$), we will assume that siblings have a correlation of 0.5 in their environmental effects throughout the rest of the simulations.

## The influence of indirect genetic effects

In order to focus attention on the impacts of non-zero indirect genetic effects, we performed simulations with the following settings: the variance of direct genetic effects normalized to 1, the variance of the environmental effect is assumed to be three times the variance of the direct genetic effect, and no correlation between direct and indirect genetic effects. As we increase the contribution of indirect genetic effects, we found that sibling PGS models produce estimates of the direct effect variance component that is attenuated compared to the estimates for the population (i.e. non-sibling) PGS models. As we showed in Fig 1, with the assumption of environmental correlations of 0.5 ($\rho_e = 0.5$) and no indirect genetic effects, the first column of results in Fig 2 are estimated accurately (see Formulas (10, 11)). However, as indirect genetic effects are introduced in Columns 2 and 3, the downward bias of the sibling model appears and becomes larger. This is consistent with empirical results from other work cited above. However, since the ratio of these estimated $R^2$ values compared with the expected direct effect component variance is smaller than 1, our results demonstrate that the sibling PGS model does not accurately recover the direct genetic effects. The figure also shows that the population (i.e. non-sibling) model cannot recover direct effects, and that the estimated effects are biased upward (as is commonly understood).

We now further consider the regression that sibling PGS model (i.e. sibling difference model) performs: $\Delta Y_j \sim \widehat{\Delta PGS}_j$. When taking the difference in sibling phenotypes, the genetic nurture effect cancels out since full siblings share the same parents (formula (6)). Whereas in estimated PGS, the transmitted genetic nurture remains different between siblings (formula (7)).

To better understand the impact of indirect genetic effects on sibling PGS analysis, we plot the ratio of sibling model $R^2$ estimated with $\widehat{\Delta PGS}_j$ compared with the $R^2$ estimated with the sibling difference in the direct genetic effect component (formula (12)). This compares the regular sibling model estimated $R^2$ with the $R^2$ of true direct effects. Note in the third box in each

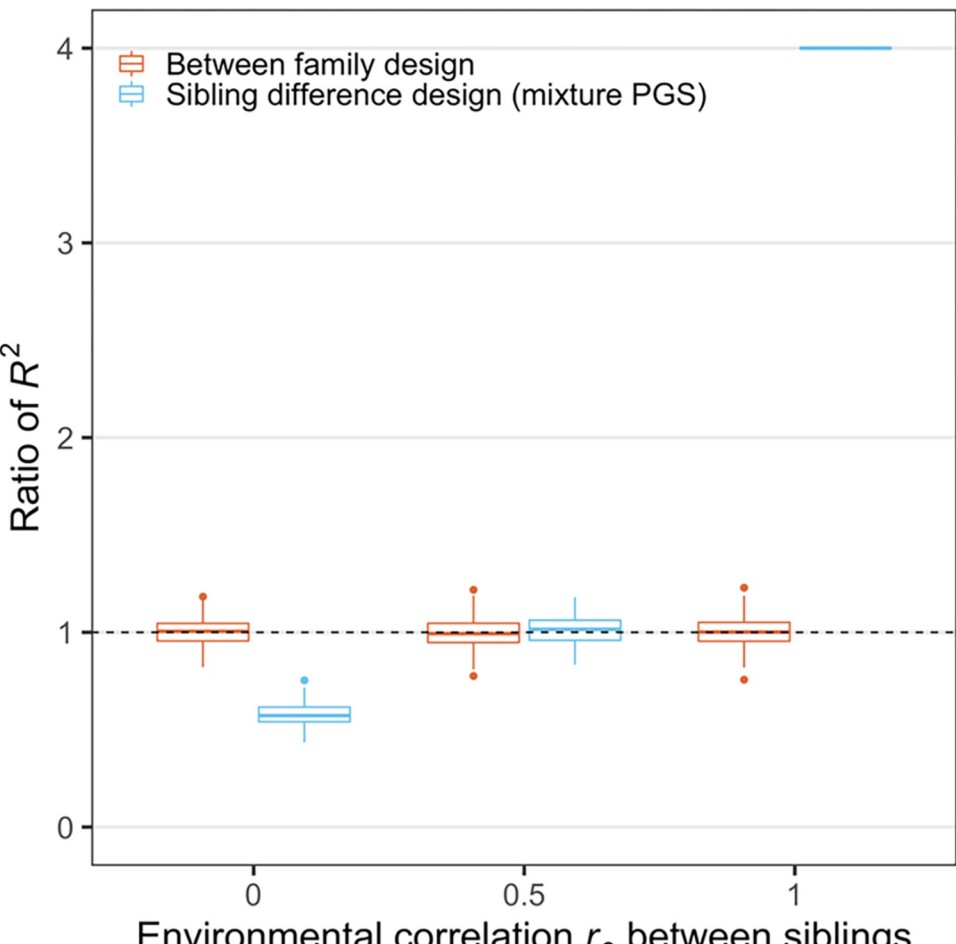

**Fig 1. The effect of environmental correlation $r_e$ between siblings on the $R^2$ of PGS regression analyses from different study designs.** y-axis shows the ratio of $R^2$ from between family design (red) or sibling difference design (blue) vs. the proportion of phenotypic variance due to the direct effect in the population. Each boxplot shows the simulation results of 200 repeats. In each repeat, we simulated the true SNP effect sizes ($\beta_{dir}$ and $\beta_{ind}$) from a bivariate normal distribution, and the environmental effect sizes for siblings from a bivariate normal distribution. We then calculated the phenotype, PGS, and run the linear regression analyses. In this figure, the PGS was calculated using ($\beta_{dir}+\beta_{ind}$). The variance of direct effect size and the variance of the environmental effect size were fixed at 1 and 3, respectively. The indirect effect size and the correlation $r_g$ between direct and indirect effect were both set to 0. When $r_e = 1$, the environmental terms for two siblings become identical, thus their phenotypic difference becomes $\Delta PGS_{dir}$ and their PGS difference is also $\Delta PGS_{dir}$ since here we set $\beta_{ind} = 0$. Thus, its $R^2$ is always 1 in each repeat whereas the proportion of the phenotypic variance by the direct effect is ¼. Therefore, the ratio is always 4 for the last setting as shown in the figure.

group, we see results below 1. Thus, the indirect genetic effect reduces the regression $R^2$ to a larger extent as its contribution to the phenotype increases. We might speculate, then, a smaller reduction in regression $R^2$ for a phenotype with moderate indirect genetic effects (e.g. asthma) compared to a phenotype with likely larger indirect genetic effects (e.g. education)—but both estimates would be affected.

## The influence of correlation between direct and indirect effects

To add an additional element to our analysis, we considered the case of non-zero correlation between indirect genetic effects and direct genetic effects. As above, we constructed

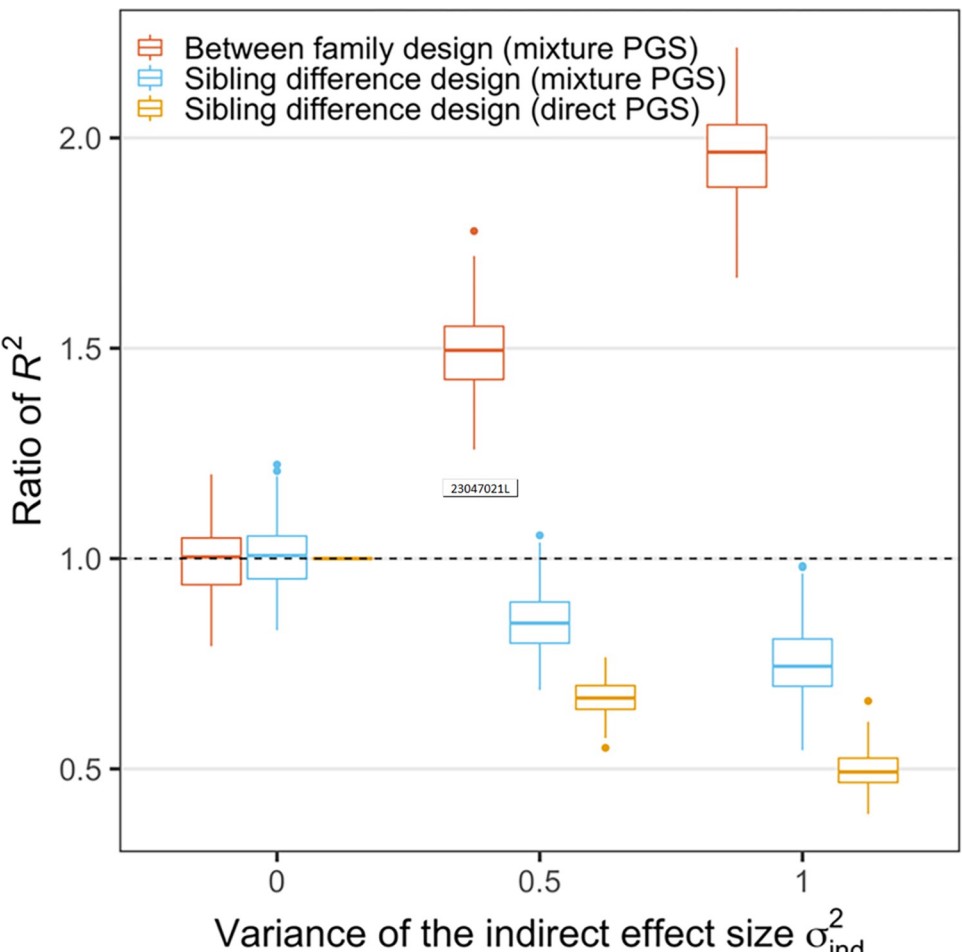

**Fig 2. The effect of indirect effect size variance on the $R^2$ of PGS regression analyses from different study designs.** y-axis shows the ratio of $R^2$ from between family design (red) or sibling difference design (blue) vs. the proportion of phenotypic variance due to the direct effect in the population. The yellow box shows the ratio of $R^2$ from the sibling difference design based on $PGS_{mix}$ vs. that in a sibling difference design based on $PGS_{dir}$. Each boxplot shows the simulation results of 200 repeats. In this figure, the variance of direct effect size and the variance of the environment effect size were fixed at 1 and 3, respectively. The correlation $r_g$ between direct and indirect effect sizes and the correlation $r_e$ between two sibling's environments were fixed at 0 and 0.5, respectively. When the indirect effect is 0 (both $\sigma_{ind}^2$ and $\beta_{ind}$ are 0), the sibling difference analyses become identical regardless of whether the PGS is computed based on ($\beta_{dir} + \beta_{ind}$) or $\beta_{dir}$, thus the yellow box is fixed at 1 when $\sigma_{ind}^2 = 0$ in this figure.

phenotypes with the variance of direct effect normalized to 1, the variance of indirect genetic effect as either a half or equal to the variance of the direct genetic effect, and the variance of environmental effect as three times the variance of the direct genetic effect. However, we now relax the assumption from above that the correlation between indirect and direct genetic effects is zero and instead varied the correlation between direct and indirect effects between -1 and 1. We found that sibling PGS models, in general, produce estimates of the of direct genetic effects that are smaller than estimates from population (i.e. non-sibling) PGS models (Fig 3). We note that previous literature viewed reductions in estimated direct effect contribution using sibling models as evidence that these models were eliminating confounds (such as indirect genetic effects), whereas our results show that sibling models actually underestimate direct genetic effects in many scenarios.

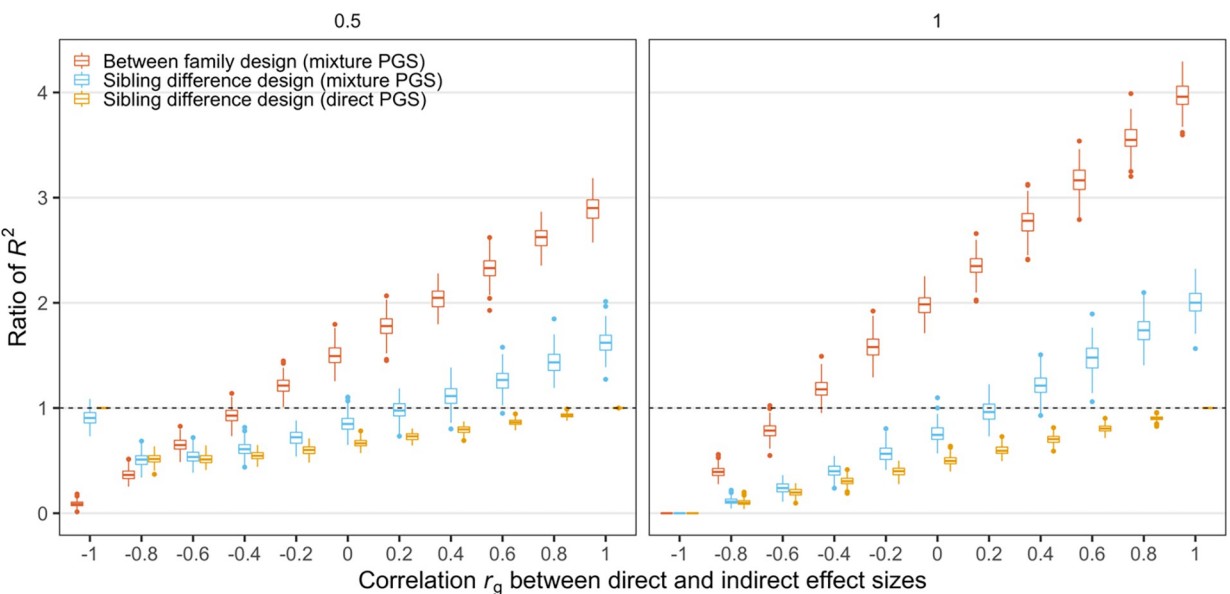

**Fig 3. The effect of correlation $r_g$ between the direct and indirect effect sizes on the $R^2$ of PGS regression analyses from different study designs.** y-axis shows the ratio of $R^2$ from between family design (red) or sibling difference design (blue) vs. the proportion of phenotypic variance due to the direct effect in the population. The yellow box shows the ratio of $R^2$ from the sibling difference design based on $PGS_{mix}$ vs. that in a sibling difference design based on $PGS_{dir}$. Each boxplot shows the simulation results of 200 repeats. In this figure, the variance of direct effect size and the variance of the environmental effect size were fixed at 1 and 3, respectively. The correlation between two sibling's environments was fixed at 0.5. The two panels correspond to the results when the variance of the indirect effect size is 0.5 and 1, respectively.

We show results for two settings that (as above) fix the variance of direct genetic effects at 1 and the variance of indirect genetic effects at 0.5 or 1. We find that increasing the correlation between direct and indirect effects increases the ratio of sibling PGS estimates and the direct effect component variance from below 1 (underestimate) to above 1 (overestimate). $R^2$ can be viewed as a function of the correlation between direct and indirect effects given specific values of other parameters, such as the variance of indirect genetic effects. Thus, the ratio of the sibling PGS estimate and the direct effect variance component (defined on the population) does not change linearly (formula (10), Appendix 3 in S1 File). Whether we allow indirect genetic effects to be modest (0.5) or large (1) and also allow correlations between the direct and indirect genetic effects, we find that it is rare that sibling analysis will accurately estimate direct genetic effects (i.e the horizontal line at 1, where the estimated $R^2$ is equal to the expected $R^2$). We note that we fine large variations in the bias: the results suggest that the estimated $R^2$ can be up to twice the size of the true $R^2$ or less than half the size of the true $R^2$, depending on the data generating process. We also note again that we have assumed in these analyses that the environmental effects are correlated at 0.5 between siblings. Otherwise, these results would "shift down", as we show in Fig 1.

## The performance of PGS regression coefficients

As another important measurement of PGS model performance, regression coefficients have also drawn attention in past research. One study showed that between-family PGS regression would yield coefficients that are biased upwards and within-family PGS regression would yield coefficients that bias downwards (Trejo & Domingue [4]). To verify this in our framework, we estimated regression coefficients in our framework on a wider value range of $r_g$ (i.e. the correlation between indirect and direct genetic effects). We also noted a divergence between our

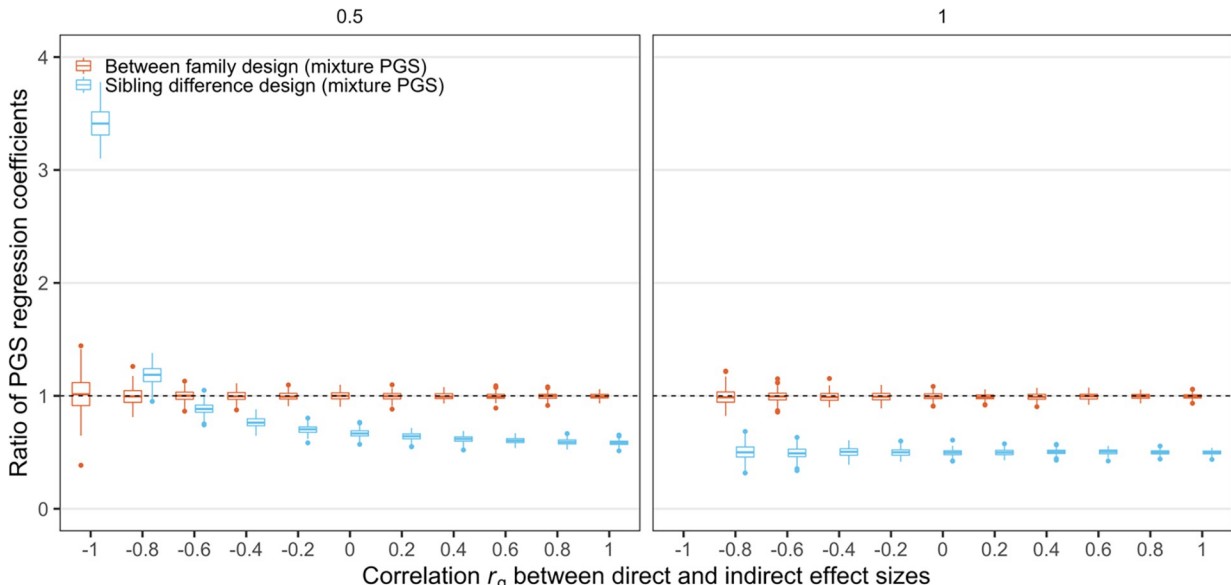

**Fig 4. The effect of correlation $r_g$ between the direct and indirect effect sizes on PGS regression coefficients from different study designs.** y-axis shows the ratio of the PGS regression coefficients from between family design (red) or sibling difference design (blue) vs. 1, which is the effect size of the $PGS_{dir}$ (see **Table 1**). We used the same simulation settings as those used in **Fig 3**. Since we set the variance of the direct effect size $\sigma^2_{dir} = 1$, when the variance of the indirect effect size is also 1 and their correlation is -1, the direct and indirect effect sizes become exactly the opposite of each other, therefore $PGS_{mix} = 0$ and the linear regression cannot be run under this scenario. Therefore, we do not include results when $\sigma^2_{ind} = 1$ and $\rho_g = -1$.

framework and previous results in terms of the standardization of PGS. Our framework does not standardize the PGS variable in the estimation, so that the between-family regression will estimate a coefficient of 1, which equals the theoretical value when the direct effect variance component is accurately recovered (Fig 4). For the sibling PGS model, with unstandardized PGS estimates, regression coefficients could be over- or under-estimated depending on the ratio of indirect and direct effect variance and the value of their correlation. Overall, the larger the ratio is, the more the sibling regression coefficients converge to a downwardly biased value. The larger the correlation is, the more the sibling regression coefficients are biased downwards (Fig 4). We also confirmed these with our derivation and rewrite Trejo and Domingue's derivation without standardization of PGS (Appendix 5 in S1 File).

## Discussion

To examine the performance of the estimated $R^2$ of sibling PGS analysis in recovering the direct genetic effect variance component from a data generating process that includes both direct and indirect genetic effects, we performed our analysis on simulated phenotypes based on genotypic data of quads in the SPARK cohort. Our analytical results demonstrated that sibling PGS analysis generally does not yield $R^2$ that accurately reflects the direct effect variance component.

In our simplified scenario where a phenotype is not impacted by indirect genetic effects, sibling PGS analysis can yield $R^2$ estimates that are biased either upward or downward, depending on the environmental correlation between siblings. In this study, we assumed a genotypic correlation of 0.5 between siblings which set a scale for the variance of the difference between their estimated PGS. More importantly, when taking the difference in sibling phenotypes, the indirect genetic effect that is shared between siblings is eliminated, leaving only the

difference in direct genetic effects and difference in environmental effects. However, PGS constructed from GWAS estimates will, in most cases, contain both direct and indirect effects. When taking a difference between siblings, the direct genetic effects and transmitted genetic nurture effects remain in the beta weights that are used to construct the PGS in downstream analysis. Given these issues, three aspects of the sibling PGS model are found to generate biases.

First, based on the composition of the sibling phenotype difference, sibling PGS regressions only retains the proportion explained by direct genetic effects and environmental effect differences. Essentially, when the phenotype is affected by indirect genetic effects, the total variance of the phenotype is reduced when using sibling analysis compared to the variance defined in population-based PGS regression. Even with accurate direct effect PGS, sibling PGS still fails to fully recover the direct effect variance component for the population.

In order to examine the impact of other factors, we turned to comparing the sibling $R^2$ estimates with the theoretical sibling $R^2$ when regressing phenotype difference on true direct PGS difference. We found that, as the contribution of indirect genetic effects increases from 0, the ratio of estimated sibling $R^2$ and the theoretical sibling $R^2$ of direct genetic effects continues to decline from its target (unbiased) value of 1. This means the indirect genetic effect component is similar to "measurement error" in this case, attenuating the direct effect estimates. Additionally, when the contribution of direct, indirect, and environmental effects is held fixed, changes in the correlation between direct and indirect genetic effect will also lead to bias in sibling $R^2$. As the correlation reduces from 1 to -1, the estimated $R^2$ is increasingly biased downwards. We label this as an "LD-like" relationship between direct and indirect genetic components, which survives sibling differencing or sibling fixed effects analysis. When compared with the direct effect variance component defined at the population level, the correlations between indirect and direct genetic effects can lead to either downward or upward bias. Similarly, we make slight adjustments on the previous results from Trejo and Domingue that focus on bias of regression coefficients (rather than $R^2$) obtained from sibling PGS analysis. Our conclusion shows that sibling analysis continues to be biased upwards or downwards in a way that depends on a combination of variances of direct genetic effects, indirect genetic effects, and their correlation.

It is important to note that our results are from a "simple" data generating process, where we assumed no assortative mating, no gene-environment interaction or correlation, and sibling genetic correlations of exactly 0.5. Adding these other elements to the framework will further complicate evaluating the performance of the sibling analysis, but, we suspect, will lead to additional biases rather than fewer. Thus, our view of the results from a relatively simple framework is that sibling analysis, coupled with conventional PGS, can rarely uncover a key target—direct genetic effects. Solving this issue will rely on dissecting each individual variant's direct and indirect effects and calculating respective PGS for direct and indirect genetic components, possibly through sibling GWAS and multi-generational analysis (Howe et al., [6]; Wu et al., [9]).

## Materials and methods

### Data

We leverage family-based genetic data from quads (2 parents, 2 children) in the SPARK (Simons Foundation Powering Autism Research for Knowledge) study (Feliciano et al [10]) in order to seed the model with realistic genetic information. Specifically, we obtained 7,026,791 SNPs from 1813 families with two parents and two full siblings. Following previous work (Huang et al. [11]), we filtered out single nucleotide polymorphisms (SNPs) with a minor allele

frequency less than 1%, with an imputation quality score less than 0.8, that are duplicated, or strand-ambiguous. Then we pruned the SNPs with linkage disequilibrium (LD) with a pairwise $r^2$ higher 0.1. From the remaining 127,310 SNPs, we randomly picked 10,000 SNPs as causal variants and simulated phenotypes based on them.

### Model specifications

We assume a data generating process for the phenotype $Y_{ij}$ that includes both direct and indirect genetic effects. Our model follows Kong et al. [1] by assuming additive separable direct ($\beta_{dir,k}$) and indirect ($\beta_{ind,k}$) genetic effects that are drawn from a bivariate normal distribution with a correlation parameter. The indirect genetic effect behaves as a family fixed effect that is shared between siblings. We denote the genotype of the $i^{th}$ child/sibling in the $j^{th}$ family as $G_{ij}$; the genotype of mother and father in the $j^{th}$ family as $G_{m,j}$ and $G_{p,j}$ respectively. We can write the model as

$$Y_{ij} = \sum_{k=1}^{M} G_{ijk}\beta_{dir,k} + \sum_{k=1}^{M} (G_{m,jk} + G_{p,jk})\beta_{ind,k} + e_{ij}, \tag{1}$$

where $e_{ij}$ is the environmental residual. We assume equal indirect paternal and indirect maternal effect, both being $\beta_{ind,k}$. We note that this parameter can also be viewed as the average indirect parental effect if maternal and paternal effects are in fact unequal (Wu et al. 2021). We also assume that the direct and indirect effect sizes of the $k^{th}$ SNP on the phenotypes follow

$$\begin{pmatrix} \beta_{dir,k} \\ \beta_{ind,k} \end{pmatrix} \sim MVN\left( \begin{pmatrix} 0 \\ 0 \end{pmatrix}, \frac{1}{M} \begin{pmatrix} \sigma_d^2 & \rho_{di}\sigma_d\sigma_i \\ \rho_{di}\sigma_d\sigma_i & \sigma_i^2 \end{pmatrix} \right). \tag{2}$$

where $\sigma_d^2$ represents the variance of the direct effect component of the phenotype; $\sigma_i^2$ represents the variance of the indirect effect component of the phenotype; and $\rho_{di}$ represents the correlation between direct and indirect genetic effect sizes; M denotes the number of causal SNPs with direct effect size $\beta_{dir,k}$ and indirect effect size $\beta_{ind,k}$. For families whose genotypes are used in the simulation, we assume that the environmental effects for the 2 children in the $j^{th}$ family also follow a bivariate normal distribution

$$\begin{pmatrix} e_{1j} \\ e_{2j} \end{pmatrix} \sim MVN\left( \begin{pmatrix} 0 \\ 0 \end{pmatrix}, \begin{pmatrix} \sigma_e^2 & \rho_e\sigma_e^2 \\ \rho_e\sigma_e^2 & \sigma_e^2 \end{pmatrix} \right), \tag{3}$$

where $\sigma_e^2$ represents the variance of environmental effects shared between 2 siblings; and $\rho_e$ represents the environmental correlation between 2 siblings. We further assume all genotypes involved are standardized.

Formula (1) can be rearranged to separate the effects of transmitted ($G_{ij}$) and non-transmitted ($N_{ij}$) alleles as

$$Y_{ij} = \sum_{k=1}^{M} G_{ijk}(\beta_{dir,k} + \beta_{ind,k}) + \sum_{k=1}^{M} N_{ijk}\beta_{ind,k} + e_{ij}. \tag{4}$$

We assumed transmitted and non-transmitted alleles to be independent (supported by genotypic data, Appendix 1 in S1 File). It is clear to see from the rearrangement that a GWAS on phenotype will capture both the true direct and indirect genetic effects. Following the conventions in the literature, we constructed the downstream PGS estimation with the theoretical GWAS estimated allelic weights $\widehat{\beta_k} = \beta_{dir,k} + \beta_{ind,k}$, assuming all causal SNPs are accurately estimated, which we denote as $\widehat{PGS}_{ij} = \sum_{k=1}^{M} G_{ijk}\widehat{\beta_k}$ (Lee et al. [12]). We obtained between-

family PGS regression coefficients $\gamma_{OLS}$ and r-squared $R^2_{OLS}$ by regressing the phenotype of one sibling from each family on their estimated PGS as

$$Y_{1j} = \gamma_{OLS}\widehat{PGS}_{1j} + e_{OLS,j}. \qquad (5)$$

Derivations on the theoretical regression coefficient and r-squared for between-family analysis are included in the Appendix 2 in S1 File. For the sibling analysis, we took the difference in the phenotype between two siblings in a family as the within-family outcome

$$\begin{aligned} \Delta Y_j &= Y_{1j} - Y_{2j} \\ &- \left( \sum_{k=1}^{M} G_{2jk}\beta_{dir,k} + (G_{m,jk} + G_{p,jk})\beta_{ind,k} + e_{2j} \right) \\ &= \sum_{k=1}^{M} (G_{1jk} - G_{2jk})\beta_{dir,k} + (e_{1j} - e_{2j}). \end{aligned} \qquad (6)$$

The shared indirect genetic effect is eliminated between siblings. We took the difference in the estimated PGS between two siblings in a family as the within-family predictor

$$\Delta\widehat{PGS}_j = \widehat{PGS}_{1j} - \widehat{PGS}_{2j} = \sum_{k=1}^{M} (G_{1jk} - G_{2jk})(\beta_{dir,k} + \beta_{ind,k}). \qquad (7)$$

Then we obtained within-family PGS regression coefficients $\gamma_\Delta$ and r-squared $R^2_\Delta$ by regressing the difference in the phenotype on the difference in the estimated PGS as

$$\Delta Y_j = \gamma_\Delta \widehat{\Delta PGS} + e_{\Delta,j}. \qquad (8)$$

When we assume siblings from the same families have a correlation of 0.5 in their genotypes, $G_{1jk}$ and $G_{2jk}$ (also supported by the genotypic data we use for simulation, details in Appendix 1 in S1 File), the within-family regression coefficients and r-squared can be derived as

$$\gamma_\Delta = \frac{\sigma_d^2 + \rho_{di}\sigma_d\sigma_i}{\sigma_d^2 + \sigma_i^2 + 2\rho\sigma_d\sigma_i} \qquad (9)$$

and

$$R^2_\Delta = \frac{(\sigma_d^2 + \rho_{di}\sigma_d\sigma_i)^2}{(\sigma_d^2 + \sigma_i^2 + 2\rho_{di}\sigma_d\sigma_i)*(\sigma_d^2 + Var(e_{1j} - e_{2j}))}. \qquad (10)$$

To quantify the performance of both analyses on recovering the direct effect component, we compared the outcome regression coefficient with 1 (as direct effect component in model (2) takes a coefficient of 1) and r-squared with the proportion of direct effect variance component defined on population base

$$h^2_{dir,OLS} = \frac{\sigma_d^2}{\sigma_d^2 + 2\sigma_i^2 + 2\rho_{di}\sigma_d\sigma_i + \sigma_e^2}. \qquad (11)$$

We also compared the outcome r-squared of sibling analysis with the proportion of direct effect variance component defined on sibling differences which allowed us to better understand the impact of the change of parameters on sibling analysis alone. That is, we performed

regression

$$\Delta Y_j = \gamma_{dir,\Delta}\left[\sum_{k=1}^{M}(G_{1jk} - G_{2jk})\beta_{dir,k}\right] + e_{dir,\Delta,j} \tag{12}$$

and obtained

$$h^2{}_{dir,\Delta} = \frac{\sigma_d{}^2}{\sigma_d{}^2 + \mathrm{Var}(e_{1j} - e_{2j})}. \tag{13}$$

(Appendix 4 in S1 File)

**Simulation.** We generated direct effect and indirect effect allelic weights for each offspring from a normal distribution from each combination of parameters and apply them to offspring's standardized genotypes. We also generated environmental effects for each offspring from a normal distribution following the parameters in each setting. By adding these components up for each offspring, we obtain their phenotypes.

*Setting 1.* From the derivations of our estimates above, we found that even in the simplest scenarios with unbiased GWAS effect sizes and genetic nurture absent, sibling analysis does not accurately estimate the variance component of direct effect. As a special case of Formula (11), here we have the population-based direct effect variance component defined as

$$h^2_{dir,OLS} = \frac{\sigma_d^2}{\sigma_d^2 + \sigma_e^2}.$$

However, $R^2$ from sibling analysis is expected to be

$$R^2_\Delta = \frac{\sigma_d^2}{\sigma_d^2 + Var(e_{1j} - e_{2j})} = \frac{\sigma_d^2}{\sigma_d^2 + 2\sigma_e^2 - 2\rho_e\sigma_e^2}.$$

Comparing these two formulas, one can see that the difference between them depends on the correlation between the environmental effect of siblings, $\rho_e$. Only when $\rho_e = 0.5$, these two quantities equal. Therefore, we designed a setting where we kept the variance of direct genetic effect constant and set indirect genetic effect to be 0. Then, the correlation between direct and indirect effect is also 0. We also set the variance of the environmental residual and set the correlation between siblings' environmental residual, $\rho_e$, to 0, 0.5, or 1. Thus, a total of 3 scenarios were examined in setting 1.

*Setting 2.* We kept the variance of direct genetic effect constant (normalized to 1) and varied the indirect genetic effect and the correlation between direct and indirect genetic effect to evaluate the influence of each factor on the sibling analysis $R^2$ when the other was held constant. Specifically, we set the variance of indirect effect to be either 0, 0.5, or 1. For each variance of indirect effect, we varied the correlation between direct and indirect effect from -1 to 1 by a step of 0.2. We also set the variance of environmental residual to be 3. A total of 33 scenarios (3 variance of indirect effect x 11 correlation between direct and indirect effect) were examined in setting 2.

## Supporting information

**S1 File.**
(DOCX)

## Acknowledgments

We thank members of the Social Genomics Working Group at University of Wisconsin for helpful comments. We are grateful to all the families participating in the Simons Foundation Powering Autism Research for Knowledge (SPARK) study.

## Author Contributions

**Conceptualization:** Jason Fletcher, Qiongshi Lu.

**Data curation:** Tianchang Li.

**Formal analysis:** Yuchang Wu, Qiongshi Lu.

**Investigation:** Qiongshi Lu.

**Methodology:** Jason Fletcher, Yuchang Wu.

**Supervision:** Qiongshi Lu.

**Visualization:** Tianchang Li.

**Writing – original draft:** Jason Fletcher, Qiongshi Lu.

**Writing – review & editing:** Jason Fletcher, Yuchang Wu, Tianchang Li, Qiongshi Lu.

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
