## [Decision Letter · Decision Letter 0]

5 May 2022

PONE-D-22-07328Interpreting Polygenic Score Effects in Sibling AnalysisPLOS ONE

Dear Dr. Fletcher,

Thank you for submitting your manuscript to PLOS ONE. After careful consideration, we feel that it has merit but does not fully meet PLOS ONE’s publication criteria as it currently stands. Therefore, we invite you to submit a revised version of the manuscript that addresses the points raised during the review process.

We look forward to receiving your revised manuscript.

Kind regards,

Heming Wang, PhD

Academic Editor

PLOS ONE

Journal Requirements:

"The authors gratefully acknowledge use of the facilities of the Center for Demography of Health and Aging at the University of Wisconsin-Madison, funded by NIA Center Grant P30 AG017266.  We thank members of the Social Genomics Working Group at University of Wisconsin for helpful comments. We are grateful to all the families participating in the Simons Foundation Powering Autism Research for Knowledge (SPARK) study."

Reviewers' comments:

Reviewer's Responses to Questions

**Comments to the Author**

1. Is the manuscript technically sound, and do the data support the conclusions?

Reviewer #1: Yes

Reviewer #2: Yes

2. Has the statistical analysis been performed appropriately and rigorously? 

Reviewer #1: Yes

Reviewer #2: Yes

3. Have the authors made all data underlying the findings in their manuscript fully available?

Reviewer #1: No

Reviewer #2: Yes

4. Is the manuscript presented in an intelligible fashion and written in standard English?

Reviewer #1: Yes

Reviewer #2: Yes

5. Review Comments to the Author

Reviewer #1: In this manuscript, the authors explored these expectations by using family (quad) data and simulations that include indirect genetic effect processes and evaluates the ability of sibling analysis to uncover direct genetic effects of polygenic scores. They pointed out that interpreting results from sibling analysis aimed at uncovering direct genetic effects should be treated with caution.

Overall, it is a solid manuscript. The authors conducted a variety of simulation scenarios to demonstrate the bias. The methodology and procedures were proposed and clearly described. The results and conclusions were clear and logical. The limitation is that the results are from a “simple” data generating processes, where they assumed no assortative mating, no gene-environment interaction or correlation, and sibling genetic correlations of exactly 0.5. But I understand that it is much more complicate in the real world.

My comments are:

1. the authors did 15 simulation repetitions, for me 15 times is kind low. I would expect that the results are generated from at least a few hundreds of repetitions.

2. In each of the boxplot, we can see that some groups have larger variations then others. Is there any explanation?

Minor comments:

1.The reference style is not consistent. For example, Introduction, second paragraph.

2. The authors should fix all the math symbols and letters. They are not showed on the pdf file.

Reviewer #2: The authors use simulated and real-world data (from parent-children quads), to determine whether within-family analyses using polygenic scores can truly separate direct and indirect genetic effects. This analysis seems especially important given that this is a claim that is often made, but not previously tested. The authors find that even under the ideal scenario (all direct, no indirect effects), within-family analyses tend to underestimate the true effect. As indirect and direct genetic effects become more correlated, estimate from within-family models become more biased in an unpredictable manner. This is an important investigation of a common assumption: that within family models are robust estimates of true direct genetic effects. It would be a welcome addition to the literature. My comments are mostly minor in hopes of clarifying some of the information provided in this manuscript.

Major points

Is this really a limitation of genetic analyses only? Could this be a broader issue of family fixed effects models in social and behavioral sciences more broadly? Is there any reason to think that the issues raised in these simulations do not also apply to phenotypic or environmental analyses within family?

Minor points:

Moving the data and methods into the main text would be helpful. I know that it helped orient me to the analyses.

The axis labels in the figures could be a bit clearer, maybe “estimated/expected”? Something that helps the reader quickly distinguish what the ratio is comparing.

Additionally, I think the figure labels could better differentiate the comparison. For example, rather than “between/between” and “within/within”, why not “between/indirect” and “within/direct”? This may better orient the reader to the fact that it is a comparison of an estimate to the true population parameter.

It might help to provide some specific examples of when we might expect different scenarios of different combinations of direct and indirect effects (e.g., height vs educational attainment).

I hope the authors take these comments in the constructive manner in which they are intended. This is a valuable analysis and important to those in the filed using polygenic scores.

6. PLOS authors have the option to publish the peer review history of their article (what does this mean?). If published, this will include your full peer review and any attached files.

Reviewer #1: **Yes: **Xiaoyin Li

Reviewer #2: No

---

## [Decision Letter · Decision Letter 1]

12 Sep 2022

PONE-D-22-07328R1Interpreting Polygenic Score Effects in Sibling AnalysisPLOS ONE

Dear Dr. Fletcher,

Thank you for submitting your manuscript to PLOS ONE. We are happy to tell you your revised manuscript in provisionally accepted. However, in order to publish your paper, please revise the format your math equations such that they will be displayed correctly.

We look forward to receiving your revised manuscript.

Kind regards,

Heming Wang, PhD

Academic Editor

PLOS ONE

Journal Requirements:

Additional Editor Comments (if provided):

Please see the note from Reviewer 1 and double check the display of math formula in your final version.

Reviewers' comments:

Reviewer's Responses to Questions

**Comments to the Author**

1. If the authors have adequately addressed your comments raised in a previous round of review and you feel that this manuscript is now acceptable for publication, you may indicate that here to bypass the “Comments to the Author” section, enter your conflict of interest statement in the “Confidential to Editor” section, and submit your "Accept" recommendation.

Reviewer #1: (No Response)

Reviewer #2: All comments have been addressed

2. Is the manuscript technically sound, and do the data support the conclusions?

Reviewer #1: Yes

Reviewer #2: Yes

3. Has the statistical analysis been performed appropriately and rigorously? 

Reviewer #1: Yes

Reviewer #2: Yes

4. Have the authors made all data underlying the findings in their manuscript fully available?

Reviewer #1: No

Reviewer #2: Yes

5. Is the manuscript presented in an intelligible fashion and written in standard English?

Reviewer #1: Yes

Reviewer #2: Yes

6. Review Comments to the Author

Reviewer #1: The authors addressed most of my comments. Some of the math formulas still did not display correctly.

Reviewer #2: (No Response)

7. PLOS authors have the option to publish the peer review history of their article (what does this mean?). If published, this will include your full peer review and any attached files.

Reviewer #1: No

Reviewer #2: No

---

## [Author Response · Author response to Decision Letter 1]

3 Feb 2023

The only issue raised was the errors in the PDF of some of the equations. The manuscript file has been replace with a PDF file by journal staff to address the errors.

---

## [Editor Report · Decision Letter 2]

10 Feb 2023

Interpreting Polygenic Score Effects in Sibling Analysis

PONE-D-22-07328R2

Dear Dr. Fletcher,

We’re pleased to inform you that your manuscript has been judged scientifically suitable for publication and will be formally accepted for publication once it meets all outstanding technical requirements.

Kind regards,

Heming Wang, PhD

Academic Editor

PLOS ONE
---

## [Editor Report · Acceptance letter]

16 Feb 2023

PONE-D-22-07328R2 

Interpreting Polygenic Score Effects in Sibling Analysis 

Dear Dr. Fletcher:

I'm pleased to inform you that your manuscript has been deemed suitable for publication in PLOS ONE. Congratulations! Your manuscript is now with our production department. 

Kind regards, 

on behalf of

Dr. Heming Wang 

Academic Editor

PLOS ONE